# Smartphone Use Side-by-Side with Burnout: Mediation of Work–Family Interaction and Loneliness

**DOI:** 10.3390/ijerph19116692

**Published:** 2022-05-30

**Authors:** Sónia P. Gonçalves, Joana Vieira dos Santos

**Affiliations:** 1Centro de Administração e Políticas Públicas, Instituto Superior de Ciências Sociais e Políticas, Universidade de Lisboa, 1300-663 Lisboa, Portugal; 2Psychology Research Centre (CIP), University of Algarve, 8005-139 Faro, Portugal; jcsantos@ualg.pt

**Keywords:** burnout, negative work–family interaction, loneliness, compulsive smartphone use

## Abstract

The main objective of this investigation is to study the relationship between compulsive smartphone use and burnout, and the potential mediating effect of negative work–family interaction and loneliness in this relationship. An online questionnaire was applied to a sample of 228 Portuguese workers, from various sectors, aged between 19 and 60 years (mean = 32.32); (standard deviation = 9.25), mostly female (64.5%; *n* = 154). The data were analyzed through descriptive and inferential statistics. The main results show that the compulsive use of the smartphone is positively and expressively related (β = 0.258; *p* < 0.001) to burnout, with compulsive users reporting more symptoms of burnout. In addition, this study shows the mediating power of negative work–family interaction and loneliness, in the relationship between compulsive smartphone use and burnout, with this effect being positive and significant (B = 0.072; 95% CI [0.026; 0.145]; B = 0.068; 95% CI [0.008; 0.141]). These results highlight the need for individuals and organizations to use smartphones with caution, as well as reinforce that companies must develop a way to prevent and treat possible risk factors associated with this phenomenon.

## 1. Introduction

Nowadays, the use of information and communications technology (ICT) is essential in the daily lives of many people across the globe. The evolution of ICT boosted the use of smartphones, which were globally connected [1], reaching 1.9 billion people in 2013, corresponding to 27% of the world population [2]. The numbers are increasing rapidly; in 2017, mobile users equaled 66% of the global population [3]. In 2020, the number of users has already reached 3 billion, and this figure is only expected to climb over the next few years [4]. More than two-thirds of the world’s population has a mobile phone, with most people now using smartphones; these are the world’s preferred choice for going online, accounting for a greater share of web traffic than all other devices combined [5].

Smartphones have made daily tasks much easier; their appeal and convenience make them highly reinforcing, which can lead to the development of a compulsive attachment [6]. Excessive smartphone use, with negative functional consequences, is frequently applied “problematic smartphone use” (PSU) [7]. According to Ting and Chen [8], problematic smartphone use is defined as a form of behavior characterized by the compulsive use of the device that results in various forms of physical, psychological, or social harm. Another expression to describe this phenomenon is “smartphone addiction”, but there is no consensus on the use due to inconsistencies in the theoretical framework and the conceptualization of behavioral addictions [7]. The excessive use of smartphones is not yet recognized as a formal clinical disorder in the Diagnostic and Statistical Manual of Mental Disorders (DSM-5) or International Classification of Diseases (ICD-10).

So, despite all the benefits that technologies have allowed, there is another world with a negative impact on users’ physical, psychological and social health, for example, diminishing the ability to think, remember, and pay attention or the decrease in psychological well-being [9], resulting in a higher level of stress, anxiety and depressive symptoms [10,11], as well as difficulties in regulating their emotions [12,13,14,15] and burnout in users [16]. Scholars argue that experiencing depression can engender psychological problems and other health-related symptoms, such as emotional and behavioral regulations [12,17]). Consequently, depression can increase people’s susceptibility to psychological, physiological, and interpersonal difficulties [18]. This problematic smartphone use, like any addictive symptom, can lead to serious consequences for the health, well-being, and life of such individuals. For example, the excessive use of new technologies, such as mobile phones, associated with the internet can cause dependence in their users, leading to gradual loneliness, isolation of the individual, and subsequent family problems [19]. In addition, the constant connection triggered by the use of smartphones, in some cases, can be associated with addiction and compulsive use [20]. For example, some estimate that the average user checks their phone around 80 times per day [21].

The evolution of new technologies allows individuals to break down existing barriers between their professional and personal lives. Problematic smartphone use is often associated with conflicts at the family level, caused by the constant connection [22]. The fact that an individual focuses too much on digital devices can lead to them neglecting face-to-face relationships, contributing to the gradual increase in loneliness.

The line of research associated with the consequences of using technology, especially smartphones, still need more investigations to better understand different types of impacts. Investing in the area and identifying the processes through which these technologies have an impact on the individual is urgently needed. Following this demand, the present study aims to characterize the use of the smartphone and its influence on burnout syndrome, as well as to explore two possible mediating processes associated with these relationships through negative work–family interaction and loneliness.

## 2. Background

### 2.1. Use of ICTs in Organizations

The strong pressure for the use of new technologies has redefined social dynamics through the implementation of new forms of interaction and collaboration between people. This fact has been extended to organizations, redefining their structures and business processes, generating results both in people and in organizations. These have a significant influence on the processes and results of organizational life [23]. From a positive point of view, the smartphone tremendously benefits the workplace by assisting communications and cooperation, while allowing flexible work and information sharing in real-time [24]. On the other hand, for workers who are severely dependent on smartphones at work, it is difficult for them to detach themselves psychologically from their work and their phones, leading to serious anxiety and stress [16].

Such technologies allow organizations to speed up processes, replication, responsiveness, and precision, leading to greater efficiency in the storage, processing, and retrieval of information. In addition, these enable employees to work remotely and be able to move anywhere, facilitating constant communication between individuals and organizations, greater flexibility in schedules, more virtual teams, and more teleworking [23]. With the constant connection, new technologies have changed the perception of individuals in relation to the concept of time and space, allowing the extension of working hours beyond the normal working day, causing individuals to work at unusual hours using their smartphones, mainly with regard to access to the electronic mailbox (e-mail) [20]. This constant connectivity means that employees can be contacted anywhere and at any time, with the individual having to work around the clock and everywhere, often feeling forced to respond immediately. This may lead the individual to neglect other areas of their life, such as family. This situation promotes in the individual the feeling that they are never free of technologies and that their personal sphere has been invaded [23,25] due to the power of these tools to break down the existing barrier between the personal and professional worlds [26].

For instance, the dependence on smartphones at work would increase workers’ perception of their job performance; however, on the negative side, it may lead to the emergence of smartphone addiction symptoms. Moreover, conscientious employees and those who perceived themselves as having good smartphone self-efficacy seem to be more likely to depend on their smartphones at work [27].

In addition, technologies allow organizations to have more remote supervision over their workers, which can put pressure on employees to perform more than one task at the same time (multitasking) and eliminate manual work, often affecting interpersonal relationships with co-workers, leading to loneliness [23]. According to Middleton and Cukier [28], the first thing that workers do when they wake up is to check their e-mails. In addition, the use of e-mail outside working hours has become normal and acceptable for them, being seen as part of their role as a worker [29]. However, this perceived normality does not prevent it from becoming a problem [30]. The introduction of technologies in the workplace appears to be associated with increased levels of stress in workers [31,32].

### 2.2. Burnout

In 1974, the American psychotherapist Herbert Freudenberger described technostress for the first time; however, Maslach and Jackson’s [33] definition is the one that gathers the greatest consensus, considering burnout to be a multidimensional construct, defined as a response to chronic work stress, characterized by three dimensions, (e.g., [33,34]):Emotional exhaustion is characterized by a loss of energy, enthusiasm, and a reduction of emotional resources, which makes the individual feel exhausted.Depersonalization is associated with the development of negative and insensitive attitudes towards customers, colleagues, or the organization, treating them as objects.Decrease in personal accomplishment, which corresponds to the tendency of professionals to not be fulfilled/satisfied with their professional performance.

This response can affect organizations and reduce productivity, and the quality of service or product and lead to a higher level of absenteeism, accidents at work, increased interpersonal conflicts, and sick leave, among others [35,36]. At the individual level, workers are emotionally and physically exhausted, without energy, with feelings of sadness, anxiety, and irritability. In addition, these symptoms can lead to physical health problems, such as ulcers, insomnia, hypertension, and, in extreme cases, can lead to the abusive consumption of psychotropic substances, triggering problems at the family and social levels [37].

### 2.3. Work–Family Interaction

The diffusion of new technologies has been affecting the balance between the different spheres of the individual’s life, by allowing the performance of several roles simultaneously [25,26]. Studies focusing on this theme have warned that greater use of ICT, especially the smartphone, is associated with a negative interaction between work and family [38].

The smartphone introduces behaviors related to work, (e.g., answering work-related emails) in the home domain. Furthermore, it is possible to answer work-related phone calls while the children are playing in the living room [39]. There is also some evidence that excessive smartphone use is associated with an increase in day-to-day work–home interference [16].

Society has undergone immense changes in recent decades, whether on economic, social, and even cultural levels, with work and family being the main contexts of contemporary society [40]. The concept of work–family, began to be defined as a negative conflict relationship between the two roles, where the performance of both was seen as incompatible, that is, the conflict caused by pressure from work or family hinders the performance of the other role [41]. With the emergence of positive psychology [42], the authors rethought the concept related to the interaction of work with the family, as being a positive interaction, that is, where the performance of a role can improve performance in another role [43]. The balance between work–family is fundamental so that there is no conflict between roles; thus, the subject must be equally dedicated and satisfied in both work and family [25].

The negative work–family interaction is closely related to less professional satisfaction, less organizational commitment, and greater absenteeism, and there are also risks associated with the health and well-being of the individual [38]. A negative interaction between work and family, or vice versa, can lead to less satisfaction with life, increasing mental health problems, such as depression, fatigue, anxiety, stress, and psychosomatic symptoms. In addition, in some cases, it can even lead to abusive behavior with the consumption of addictive substances [38].

In the work–family interaction, burnout has been identified as an important consequence [44,45]. According to Mete and colleagues [46], people who are not happy with their jobs and workplace environment may have conflicts in their families. The authors also suggest that the continuity of conflict may trigger more burnout.

### 2.4. Loneliness

Loneliness has become a silent plague [47]. This trend seems to have become prevalent across the globe, with 40% of Americans reporting that they are feeling lonelier and more isolated than ever [48]. Those with a high level of loneliness tend to use smartphone communication more, while being reluctant to engage in face-to-face interaction [47].

Loneliness and isolation grow behind the scenes, when the individual neglects face-to-face contact with friends and family, often taking refuge in an identity created in the virtual world [19].

There are several definitions for loneliness, but there are three points in common: perceived deficits in the individual’s social relationships—subjective experience; the individual does not have the type of relationship they want or the quantity/quality of it; an unpleasant experience that can involve negative consequences [49,50].

In this research, loneliness is defined as a psychological state, that is, it is seen as a feeling that differs from individual to individual, from time and circumstances of one’s own life [51].

In several studies, loneliness has appeared to be associated with poor physical and mental health [49,52] and is often linked to depression, hostility, suicide, and psychosomatic diseases, among others [50,53]. Loneliness has been estimated to shorten a person’s life by 15 years, equivalent in impact to being obese or smoking 15 cigarettes per day [54].

### 2.5. Conceptualization and Hypotheses

Smartphones have become inevitable in people’s daily activities, both personal and professional. In one way, this usage has a lot of potential for communication and interactions, in another way the constant use of smartphones increases the demands on the individual’s own capacity to set limits for use and accessibility, elevating users’ stress [55]. Having sleep repeatedly interrupted can have direct effects on recovery and health. According to Harper [56], technostress can lead to burnout symptoms. Van Eck [57] states that, through the results obtained from his study with professionals and users in the IT area, technostress can lead to high levels of stress and, in turn, to burnout.

Some studies on work-related smartphone use after work have considered it a job demand based on the JD-R theory [58]. Since work-related smartphone use after work brings about another job, workers are likely to experience burnout because they have to devote more time and energy to their jobs in their non-work hours [59].

As a result of studies that point to burnout as a potential negative consequence of technologies, we formulate the following hypothesis: 

**Hypothesis** **1** **(H1).**
*The higher the level of smartphone use, the higher the level of burnout.*


Derks and Bakker [16], in a study about the impact of daily recovery experiences on daily work–home interference (WHI) and daily burnout symptoms within a group of smartphone users, found strong evidence that smartphone use during after-work hours has a significant impact on the work–home interface. Additionally, the intensive smartphone user’s experience has a stronger relation between WHI and burnout levels. Employees who have the habit of always staying connected to their work through their smartphones have more difficulty psychologically detaching from work. In other words, work–home interference impedes the recovery process.

The costs of high work–home interference in burnout exhaustion are disproportionally loaded on the intensive smartphone user [16]. In this way, we formulate as

**Hypothesis** **2** **(H2).**
*The relationship between smartphone use and burnout is mediated by the negative work–family interaction.*


Despite the negative work–family interaction, earlier studies, (e.g., [60]) show a positive association between loneliness and problematic use of a smartphone. Loneliness has also been found to be significantly associated with Internet addiction [61]. Engelberg and Sjöberg [62] found that lonely people with poorer social skills tend to have more frequent use of the Internet. From here, we predict

**Hypothesis** **3** **(H3).**
*The relationship between smartphone use and burnout is mediated by loneliness.*


The main objective of this investigation is to study the relationship between compulsive smartphone use and burnout. Considering the previous literature presented it is also objective to analyze the potential mediating effect of negative work–family interaction and loneliness in the relationship between compulsive smartphone use and burnout.

## 3. Materials and Methods

### 3.1. Sample

A total of 228 Portuguese workers participated in the study, 67.5% (*n* = 154) were women, aged between 19 and 60 years, with an average age of 32 (SD = 9.25). The majority of individuals had a higher education level, considering that more than 80% of the participants (*n* = 184) had a degree. Regarding marital status, the majority of the sample was single (*n* = 132; 57.9%). Only 80 (35.1%) individuals had children. With regard to the work context, participants had diverse professions and different experience levels, with an emphasis on individuals with 6 to 24 months of experience in their current professional sector (*n* = 65; 28.5%) and individuals with more than 10 years of experience (*n* = 61; 26.8%).

### 3.2. Materials

Smartphone use was assessed using the Compulsive Usage of Mobile Phones Scale [55], one single factor scale. This scale seeks to characterize the compulsive use of smartphones through a set of 13 questions, (e.g., “I can’t concentrate on work because of the phone/smartphone” and “I try not to use the phone/smartphone as often, but without success”), measured on a 5-point Likert scale, ranging from 1 (“Strongly Disagree”) to 5 (“Strongly Agree”) (α = 0.84). The scale was translated into Portuguese as part of the prospective study using the translation and back-translation method.

To measure burnout, the MBI—Maslach Burnout Inventory [63] (Portuguese version [64]) was used. It consists of 22 items, (e.g., “I feel tired when I get up to go to work”) answered on a 7-point Likert scale, ranging from 1 (“Never”) to 7 (“Every day”) (α = 0.75) This scale can be used as a one single factor scale.

Negative work–family interaction was assessed using the negative work–family interaction sub-scale of SWING—Survey Work–Home Interaction Nijmegen, created by Geurts and collaborators [65] and adapted to Portuguese by Pereira and collaborators [40]. This subscale is composed of 8 items, (e.g., “Having to cancel programs with family, friends or spouse due to work commitment), answered on a 4-point Likert scale, ranging from 1 (“Never”) to 4 (“Always”) (α = 0.92).

The loneliness construct was assessed using the UCLA Loneliness Scale, one single factor scale, developed by Russell, Peplan, and Cutrona [66], and validated for the Portuguese population by Neto [51]. It consists of 17 items, (e.g., “There is no one to turn to”). It is a 4-point Likert scale, ranging from 1 (“Never”) to 4 (“Often”) (α = 0.91).

### 3.3. Procedure

#### 3.3.1. Data Collection Procedure

Data was collected using a self-fulfillment online questionnaire, giving the participants greater autonomy in their responses. The questionnaires were distributed through social media, such as Facebook, where people were asked to answer and share the survey; later, in order to obtain a more heterogeneous population in the study, we opted to share the questionnaire through LinkedIn, as it is a social network aimed at professionals, whose objective is sharing and searching for professional experiences. It was shared on several pages and sent individually to selected products from various products that agreed to collaborate in the reward of data, which ended up being received with the snowball effect.

The study was conducted according to the guidelines of the Declaration of Helsinki and the ethical principles according to the Code of Ethics of the Order of Portuguese Psychologists regarding research with human beings: information on the objective, risks, and benefits of the study, protection of personal data and guarantees of confidentiality and the possibility of abandoning the study in any of its stages. Ethical review and approval were not required for the study on human participants in accordance with the local legislation and institutional requirements. Informed consent to participate in this study was provided by the participants.

#### 3.3.2. Data Analysis Procedure

After collecting the questionnaires, data were introduced and analyzed using the Statistical Package for the Social Sciences (SPSS, version 23). Only questionnaires that had 80% of the questions filled out were considered valid for data analysis.

In order to analyze the data collected, the internal properties of the scales were analyzed, and descriptive statistics were used to characterize the sample. Harman’s single factor score was used to test common method bias [67], revealing a total variance for a single factor is less than 50%, it suggests that common method bias does not affect our data, hence the results.

In order to test the proposed research hypotheses, inferential statistics were performed and, with the process (Andrew F. Hayes, version 2016), bootstrapping macro (Andrew F. Hayes version 2016), using model 4, designed to test mediation models in SPSS, as in this specific case, with two mediating variables [68]. Additionally, in order to understand the magnitude of the effect between variables, Cohen’s d was tested [69].

## 4. Results

### 4.1. Characterization of Smartphone Usage

Regarding the characterization of the sample in relation to smartphones, most individuals only had one smartphone (*n* = 228; 86.84%), with 36 individuals (15.8%) having two smartphones and the rest having more than two.

For the use of the smartphone in the work context, most participants use their device for work matters (76.8%), with around 66% having access to work email on their smartphone.

### 4.2. Characterization of Variables in Study

Analysing Table 1, burnout (M = 2.399; SD = 0.765), compulsive smartphone use (M = 2.438; SD = 0.617), loneliness (M = 1.791; SD = 0.477) and Negative work–family interaction (M = 2.095; SD = 0.664), have scores below average. The analysis of Cronbach’s alpha values, on the diagonal of Table 1, reveals that the indicators have good internal consistency.

In Table 1, we can see that all variables are significantly correlated. Compulsive smartphone use correlates significantly and positively with burnout (r = 0.258; *p* < 0.001). Table 1 also verifies that compulsive smartphone use is positively correlated with loneliness and negative work–family interaction (r = 0.148; *p* < 0.05 and r = 0.311; *p* < 0.001). With regard to the negative work–family interaction, this is positively correlated with the remaining variables. The loneliness variable correlates significantly with all the variables studied, highlighting its correlation with burnout (r = 0.452; *p* < 0.001).

### 4.3. Hypothesis

All the pre-established hypotheses were initially analyzed using simple linear regression to interpret the predictive effect of compulsive smartphone use (X) on burnout (Y) and its dimensions. Subsequently, the mediating effect of negative work–family interaction (M1) and loneliness (M2) in the relationship between compulsive smartphone use (X) and burnout (Y) is analyzed through PROCESS, model 4, to perform this test in an integrated way, including the control of sociodemographic variables (the variability of the sample was controlled, through the variables of gender, marital status, education, parenting, and professional experience), which show no statistically significant differences.

#### 4.3.1. Analyze the Predictive Effect of Compulsive Smartphone Use on Burnout

In order to test the predictive value of compulsive smartphone use in burnout, we utilized linear regression. Compulsive smartphone use has effects on burnout; as illustrated in Table 2, the results show that the model is significant (F(1, 226) = 16,087; *p* < 0.001). We also concluded that compulsive smartphone use predicts approximately 6% of the total variance of burnout (R^2^a = 0.062). The relationship between the variables is positive and significant, which shows that the greater the compulsive smartphone use, the higher the burnout rate (β = 0.258; *p* < 0.001), confirming hypothesis H1.

#### 4.3.2. Modelling Tests with Mediation

In order to test Hypotheses 2 and 3, the mediation relationship between the variables under investigation was conducted.

Testing the effect of mediations of negative work–family interaction and loneliness in the relationship between compulsive use of smartphones and burnout.

Hypotheses 2 and 3 predict the existence of a negative relationship between work–family interaction and loneliness in the relationship between compulsive smartphone use and burnout, respectively. Thus, we consider compulsive smartphone use as a predictor variable (X), burnout as a dependent variable (Y), and negative work–family interaction (M1) and loneliness (M2) as mediating variables.

This model is important to study the effect of variable X on Y, in the presence of other variables that mediate this relationship (M1 and M2).

First, to analyze the hypotheses of double mediation, we analyzed the effects of simple interaction, through linear regressions. Analysing the total effect of the mediation model, between compulsive smartphone use and burnout, we recall that the effect is significant, positive and small (B = 0.320; t = 4.011; d = 0.046; *p* < 0.001), revealing that compulsive smartphone use explains about 7% of the burnout variance (R^2^ = 0.066; F(1, 226) = 16.087; *p* < 0.001).

In the test of the effect between the mediating variables (negative work–family interaction (M1) and loneliness (M2)) and the independent variable (compulsive smartphone use (X)), it is possible to conclude that compulsive smartphone use predicts about 10% of the variance of the negative work–family interaction (R^2^ = 0.097; F(1, 226) = 24.236; *p* < 0.001), with a positive, significant and strong effect (B = 0.335; t = 4.923; d = 0.906; *p* < 0.001). Regarding the effect of compulsive smartphone use on loneliness, it predicts just over 2% of the variance (R^2^ = 0.022; F(1, 226) = 5.053; *p* = 0.026), which is significant, positive and moderate effect (B = 0.114; t = 2.250; d = 0.456 *p* = 0.026). The two regressions presented previously predict the values of interactions a1 and a2, shown in Figure 1.

In order to analyze the values referring to the effect of the predictors, namely the two mediators (negative work–family interaction (M1) and loneliness (M2)) and the independent variable (compulsive smartphone use (X)), a multiple linear regression was performed. This model explains 27% of the variation in the burnout variable (R^2^ = 0.270; F(3, 224) = 27.595; *p* < 0.001). Analyzing this regression, we observed that negative work–family interaction, loneliness and compulsive smartphone use are positively and significantly correlated with burnout, namely with a B = 0.215, t = 2.967, *p* = 0.003 and B = 0.593, t = 6.106, *p* < 0.001, referring to the value of b1 and b2 (see Figure 1a) and finally, B = 0.180, t = 2.412, *p* = 0.033, corresponding to the value of the direct effect (c’).

Regarding the conditional indirect effects of compulsive smartphone use in burnout through negative work–family interaction and loneliness, both are significant (B = 0.072; 95% CI [0.026; 0.145]; B = 0.068; 95% CI [0.008; 0.141]), with the total indirect effect of these mediations also being significant and positive (B = 0.140; 95% CI [0.056; 0.231]).

## 5. Discussion

The aim of this study was to analyze the association between smartphone overuse and burnout, hypothesizing that smartphone overuse is associated with the development of burnout through increased loneliness and work–family conflict. This model is based on the perspective of technostress, in which the excessive use of new technologies causes stress [23]; thus, we try to understand if this can be positively and significantly related to burnout (Hypothesis 1). According to the observed results, it is indeed possible to verify that compulsive smartphone use is positive and significantly related to the levels of burnout; that is, individuals with greater compulsive smartphone use tend to have a greater risk of experiencing burnout. This fact, as previously mentioned, may be related to the stress caused by the excessive use of ICT, promoting feelings of anxiety, depression, and stress, which are associated with the phenomenon of burnout [13,23,38].

Analyzing the mediation effect of negative work–family interaction in the relationship between compulsive smartphone use and burnout confirms Hypothesis 2, that negative work–family interaction mediates this relationship. This mediation is interpreted as follows: the higher the scores for compulsive smartphone use, the greater the negative work–family interaction; that is, for each point in compulsive use the interaction increases by 0.335 points, which in turn is related to the levels of burnout, that is, for each point added to the negative work–family interaction, the burnout scores increase by 0.215 points. These data show that when the relationship between compulsive smartphone use and burnout is explained in the presence of negative work–family interaction, the model assumes a greater explanation value, adding 0.072 points to the direct effect of 0.180.

The data presented in the previous paragraph are sustained in the perspective that the diffusion of new technologies has been affecting the balance between the different parts of the individual’s life by allowing them to perform several roles simultaneously [25,26]. In relation to the negative work–family interaction, this can lead to lower life satisfaction, increasing mental health problems, such as depression, fatigue, anxiety, stress, and psychosomatic symptoms, which can promote the appearance of burnout [38,45].

Observing the mediation effect proposed in Hypothesis 3, in which the relationship between compulsive smartphone use and burnout is expected to be mediated by loneliness, the results confirm this hypothesis. Thus, it is possible to infer that the relationship between compulsive smartphone use and burnout is best explained in the presence of loneliness, adding 0.068 points to the direct effect of 0.180 points; that is, higher levels of compulsive smartphone use led to greater feelings of loneliness, which culminates in higher burnout levels.

In the current literature, several other studies have identified that the use of technologies, in particular their excessive use, has a negative effect on the way we interact with others, often being associated with increasing loneliness [70,71].

When analyzing the completed model, it is possible to conclude that there is an effect of compulsive smartphone use in burnout without the presence of mediators, increasing the levels of burnout by 0.320 points, for each point increased in compulsive smartphone use, this is called the total effect of the model. When tested in the presence of mediating variables, this effect, called the direct effect, decreases to 0.180 points, that is, for each unit increase in compulsive smartphone use, the burnout levels increase by 0.180 points. The remaining effect is explained by the presence of the two mediating variables, called the total indirect effect, of 0.140 points when the model is explained by the indirect effects of the mediations.

This shows that negative work–family interaction and loneliness are two predictors of burnout and that both can be consequences of compulsive smartphone use, thus having mediating roles in this process. In addition, it warns organizations not to ignore the disorders associated with this compulsive use [13].

When analyzing the use of mobile technologies for non-professional matters, several questions arise regarding their prohibition in the workplace. Highlighted as one of the main distracting factors in work activities, although the employers’ prohibition on the use of personal smartphones in the workplace seems to be a solution to reduce its negative impacts, it shows adverse effects, such as the violation of the rules by employees, motivated by the personal desire for dependence on technology. According to Cappellozza et al. [72], the total restriction of private smartphones, on the part of organizations, can lead to dissatisfaction in the workplace, due to the decrease in positive feelings provided by their use. Thus, these measures can become a problem for organizations, as this use is often associated with technological dependence, to which employees may already be subject.

External control of the organization, in relation to the excessive use of ICT by its employees, is necessary considering that they, by themselves, may experience difficulties in adapting their use, derived from their addictive behavior [72]. A study by Cappellozza et al. [72] clarified that there are individual factors linked to technological dependence that motivate the use of personal technologies at work, in addition, the main antecedent that leads to this behavior is linked to the loss of impulse control for the use of ICT. The authors concluded that technological dependence directly interferes with individuals’ professional performance. Derived from the constant interruption of work tasks due to a lack of impulse control, leads to a reduction in individual productivity. Thus, supervision can be a good ally in the identification of cases of excessive smartphone use and the detection of low professional motivation caused by such use, in order to promote the motivation of employees and reduce the possibility of dismissal [73].

Organizations should be aware when they intend to adopt mobile technologies for workers that this use in professional terms transcends barriers, which can have negative implications. This can result in increased dissatisfaction, decreased productivity, and increased worker turnover, which is why it is important to establish limits on this use [25,73]. Examples of defining limitations are countries, such as France, that have implemented the “right to disconnect”.

Some companies have adopted rules in order to avoid prolonging the work, such as Volkswagen and Deutsche Telekom, which in 2011 decided to shut down their computer servers in the period between 6:15 pm and 7:00 am. In some cases, a code of conduct is agreed upon between the employer and employees that includes a rest period in which the employee must not be connected. In France, being a law, if the company does not proceed with this agreement, the worker can evoke their right to remain offline as a justification for this non-connectivity [74]. According to the Society of Human Resources Management [75], workers in France have to turn off their work phones after 6 pm, so as to not be contacted via calls, messages, or email, and must ignore all connectivity linked to work after that time; they cannot be penalized for such action.

According to a report by Deloitte (April, 2015), 71% of workers have accessed their work email at night or during holidays, either by obligation or willingly. The truth is that this number tends to increase alongside the technological evolutions that have happened in recent years; in the previous year, about eight in ten Europeans used the internet without a cell phone [74]. A study developed by the Portuguese Association of Occupational Health Psychology involving 38,719 workers (March, 2017) revealed that the constant connection is a public health problem that has been increasing, showing that from 2008 to 2013 Portuguese workers have shown signs of exhaustion, increasing from 9 to 15%. Regarding stress, in some periods, the levels have doubled from 36 to 62%; in addition, most respondents (78%) want to change jobs in the next five years, which proves the turnover cited by other studies, (e.g., [25,73]) consequently for these associations.

It is suggested that organizations, to avoid the development of burnout, define internal conduct that prohibits employees from responding to emails or calls outside their working hours, which could lead to serious implications for the organization.

Based on previous investigations, (e.g., [76,77]), the implementation of sports groups seems to be crucial, as these can prevent or delay the appearance of work disturbances. During the practice of sports, the individual is removed from their smartphone and such activity may reduce loneliness and loneliness and increase affinity, which can contribute to the reduction of cases of burnout.

In short, we intend for there to be greater awareness of both organizations and government entities on this issue, and for organizations and unions to put pressure on governments to implement laws that control the extension of work through technologies. As in France, a law should be implemented in Portugal that would give employees the right to disconnect, without being afraid of being penalized for this.

The constant online connection is a general health care issue, and the constant work demands reinforce it. The technology linked worldwide improves our lives, but also it is important to take care of individuals, as employees and persons. For example, outdoor activities, (e.g., football, running) are beneficent to individuals and for their sense of belonging within their team. Another example is family-friendly activities, (e.g., company group health plan, gym membership, flexible working arrangements, wellness strategies, and fairness at work) are positively related to the well-being of employees, (e.g., [78]). Both kinds of strategies promote a protective barrier from burnout.

One of the main limitations of this study is its sample size, which does not allow generalized statistical inferences to be made about the wider population. Another limitation is related to self-reporting measures, which can lead to possible biases, derived from social desirability, random responses, counterfeiting, and response style [79]. In addition, it is a correlational and cross-sectional study, which does not allow the inference of causality and may have associated effects of the common method [67].

As suggestions for future research in this area, we propose the use of a concrete quantitative measure of the actual use of smartphones. In this investigation, we tried to use battery costs, but because it is an unreliable and uneven sample method among the participants, it was removed from the study. As such, we propose the use of new methodologies that control what type of activities the individual uses his equipment and how much time they spend on these activities. Apple [80] recently launched a feature on its system (IOS 12.0.1) called “screen time” that controls the use of the smartphone, generating weekly reports with the time of use per app and respective category, the number of notifications received, and the number of interactions with the device. We suggest using the negative work–family interaction dimension, considering the impact that family problems can bring to the professional sphere. One factor that can bring more value to this type of study is to understand what leads individuals to use personal technologies within the workplace and what leads them to use work technologies outside working hours.

On the other hand, in the future, it would be interesting to better understand the effect of other types of independent predictors, such as psychological distress, time spent using the smartphone overnight, and use of image-based social networks on problematic smartphone use [81]. Another interesting point of view to better understand in future research could be the way as problematic smartphone use from significant other people, such as parents or team leaders, could affect the smartphone use of the individual [82].

In addition to considering future research, in the model other variables could be integrated, such as positive political work environment or organizational support and their impact to decrease [83]. Organizations can protect employees’ mental health by actively encouraging psychological detachment from work and by helping manage work–family conciliation (for the relationship between compulsive smartphone use and burnout, and the potential mediating effect of negative work–family interaction and loneliness in this relationship [84]. A recent study showed that the medical staff (Huelva) who had been in contact with situations of SARS-CoV-2 in their work environment presented worse indicators of mental health and greater negative interaction of work over family than those who had not been in contact with these situations [85]. So, in future research, we consider better understanding the positive impact of the work environment and support in decreasing compulsive and maladaptive smartphone use.

## 6. Conclusions

The added value of this study is based on a better understanding of the processes through which the use of smartphones impacts the worker’s health, more specifically in terms of burnout. This investigation presents a vital contribution to the study of the impacts of excessive smartphone use in the work context, allowing a first step for the reflection and development of strategies that promote the minimization of technological paradoxes in the organizational world.

## Figures and Tables

**Figure 1 ijerph-19-06692-f001:**
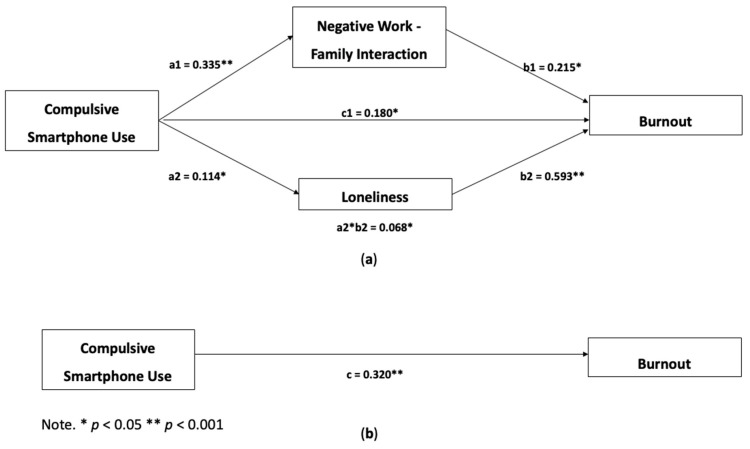
Duple mediation model (**a**) and Direct effect (**b**).

**Table 1 ijerph-19-06692-t001:** Descriptive Statistics and correlations.

	1	2	3	4
*1*. *Compulsive smartphone use* *2*. *Burnout* *3*. *Loneliness* *4*. *Negative Work–Family Interaction*	(0.84)	0.258 ***	0.148 *	0.311 ***
	*-*	(0.75)	0.452 ***	0.352 ***
	*-*	-	(0.91)	0.326 ***
	*-*	-	-	(0.92)
Mean	2.438	2.399	1.791	2.095
Standard Deviation	0.617	0.765	0.477	0.664

Note. * *p* < 0,05 *** *p* < 0.001; in the diagonal in parenthesis are the internal consistency values (Cronbach’s Alpha).

**Table 2 ijerph-19-06692-t002:** Linear regression of smartphone use and burnout.

	Standardized Beta Coefficients				ANOVA		
	Beta	t	Sig.	F	df	Sig.	R^2^ Adjusted
Compulsive smartphone use	0.320	4.011	0.000	16.087	1	0.000	0.062

Dependent variable: *Burnout.*

## Data Availability

The data that support the findings of this study are available on request from the corresponding author, Sónia P. Gonçalves, spgoncalves@iscsp.ulisboa.pt.

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
