# Peer review of "Smartphone Use Side-by-Side with Burnout: Mediation of Work–Family Interaction and Loneliness"

_ijerph, 2022, doi:10.3390/ijerph19116692_

Round 1

Reviewer 1 Report

This paper studies the relationship between compulsive smartphone use and burnout, including other variables such as work-family interaction and loneliness.

Suggestions and questions (answers can/should be used to improve the paper):
1. Who do the authors mean by "The data were through descriptive and inferential statistics."?
2. Consider "This obligation must highlight...", what obligation?
3. The introduction section is too big. It could split into "Introduction" and "Background" sections.
4. What is "trait smartphone use"?
5. Lines 223-225 have the same text as the abstract. Detail the objective in the introduction section.
6. Was this study approved by Ethics Committee? This is completely missing in the manuscript.
7. There are three sections named "Conceptualization and hypotheses" 1. - 1.5.; 2. - 1.1. and 1.2.
8. In the abstract, the authors say "An online questionnaire was developed...", but the authors used scales already existing. Why "was developed"?
9. What were the selection criteria for participants?
10. Conclusion section is too big. Some texts should be repositioned to the discussion section (e.g., limitations, future work and, mainly, discussion with citations -  there should be no citations in the conclusion). Discussion could have different subsections.
11. The authors argue "The line of research associated with the consequences of using technology, especially smartphones, remains scarce.", but if the smartphone is currently the main device to access Internet, studies focused on Internet addiction could be cited, so they are not scarce. Unpack what you mean in such a statement. Perhaps studies investigating exactly burnout, work-family interaction or loneliness. My point is because this sentence is too 'strong'.

Specific comments:
- line 140: ICT’ - remove '
- line 157: Check the text "...and satisfied in both, both at work and in the family"
- line 167: "...and colleagues [46] [add comma here]"~
- line 168: "Authors also focuses..." -> focus
- This sentence should be improved/rephrased: "Employees who have the habit of staying always connected to their work by means of their smartphones make it very hard, if not impossible, for themselves to psychologically detach from work at the same time." - if not impossible??
- In the discussion -> The aim of this study WAS
...
I strongly recommend an [English] review in the text, looking for typos, grammar errors, and points to be improved.

Reviewer 2 Report

I found this a very interesting study of the relationship between smartphone use and burnout with negative work-family interaction and loneliness acting as mediating variables. I have a couple of questions about the statistical analyses. On page 7, mention is made of controlling for sociodemographic variables, including gender, marital status, etc with no statistically significant differences being found in the model. My question is whether these differences were measured with respect to the outcome variable (burnout). Were there any differences in smartphone use or either of the mediating variables with respect to these demographic measures? I think the authors could be clearer with respect to this part of the manuscript. Also, did the authors conclude there was partial mediation or complete mediation? From what I can tell, the authors are only claiming partial mediation, but again this could be clarified in the manuscript. Finally, perhaps the authors could speculate as to whether other relevant variables not included in the study could change the results. For example, omitted variables could lead to a different interpretation if they were included. I do not know much about the subject of the smartphone use, but perhaps the authors could list this as a potential limitation in the Conclusions section and speculate as to what omitted variables might have been included. 

Concerning the use of English, I found the manuscript to be generally well-written, but in various places odd phrases were used or the English wasn't up to standard, so further proofreading is needed. 

Reviewer 3 Report

Comments to Authors 

            The current study has showed that 1) the compulsive use of the smartphone is positively and expressively related (β = 0.258; p <0.001) to burnout, with compulsive users reporting more symptoms of burnout; 2) the mediating power of negative work-family interaction and loneliness, in the relationship between compulsive smartphone use and burnout, with this effect being positive and significant.

           Authors are kindly requested to emphasize the current concepts about these issues in the context of recent knowledge and the available literature. This articles should be quoted in the References list.

           Organisations can protect employees' mental health by actively encouraging psychological detachment from work and by help managing work-family interface (for the relationship between compulsive  smartphone use and burnout, and the potential mediating effect of negative work-family interaction and loneliness in this relationship), slatentizing with the Covid 19 [1]. Infact, the medical staff of Huelva who had been in contact with situations of SARS-CoV-2 in their work environment presented worse indicators of mental health and greater negative interaction of work over family than those who had not been in contact with these situations [2].

             I think that this findings too emphasise the importance of a clear organisational policy regarding smartphone use during after-work hours.

References

  1. A nationwide cross-sectional study of workers' mental health during the COVID-19 pandemic: Impact of changes in working conditions, financial hardships, psychological detachment from work and work-family interface. BMC Psychol. 2022;10(1):73. Published 2022 Mar 18. doi:10.1186/s40359-022-00783-y.
  2. Work-Family Interaction, Self-Perceived Mental Health and Burnout in Specialized Physicians of Huelva (Spain): A Study Conducted during the SARS-CoV-2 Pandemic. Int J Environ Res Public Health. 2022;19(6):3717. Published 2022 Mar 21. doi:10.3390/ijerph19063717.

Round 2

Reviewer 1 Report

The authors improved the manuscript, and most of my concerns were addressed. However, one comment remains:

My comment 6. Was this study approved by Ethics Committee? This is completely missing in the manuscript.

Authors' answer: Information was added in procedure subsection: The study was conducted according to the guidelines of the Declaration of Helsinki and the ethical principles according to the Code of Ethics of the Order of Portuguese Psychologists regarding research with human beings: information on the objective, risks, and benefits of the study, protection of personal data and guarantees of confidentiality and the possibility of abandoning the study in any of its stages.

My reply: This does not answer my question. To be conducted, shouldn't this study have been approved by an ethics committee? Please, see section "Institutional Review Board Statement" (from the MDPI template) in your paper.

Author Response

Thank you so much for your feedback.

Considering the required we added a sentence on the procedure: Ethical review and approval was not required for the study on human participants in accordance with the local legislation and institutional requirements. Informed consent to participate in this study was provided by the participants.

In attach follows the new version of the manuscript, signalized in blue this sentence.
